# The History and Future of Basic and Translational Cell-Free DNA Research at a Glance

**DOI:** 10.3390/diagnostics12051192

**Published:** 2022-05-10

**Authors:** Peter B. Gahan, Heidi Schwarzenbach, Philippe Anker

**Affiliations:** 1Fondazione “Enrico Puccinelli” Onlus, 06126 Perugia, Italy; 2Department of Gynecology, University Medical Center Hamburg-Eppendorf, 20246 Hamburg, Germany; hschwarzenbach@me.com; 3Independent Researcher, 74160 Beaumont, France; philippeanker@gmail.com

**Keywords:** DNA, history, cancer, clinical medicine, methodology

## Abstract

We discuss the early history of the structure of DNA and its involvement in gene structure as well as its mobility in and between cells and between tissues in the form of circulating cell-free DNA (cfDNA). This is followed by a view of the present status of the studies on cfDNA and clinical applications of circulating cell-free tumor DNA (ctDNA). The future developments and roles of ctDNA are also considered.

## 1. Introduction

Although Mendel and Métais [1] described the presence of nucleic acids in blood from healthy donors, pregnant women and clinical patients in 1948, this study was largely forgotten until in 1973 the paper of Koffler et al. [2] described raised DNA levels in the blood of lupus erythematosus patients. At the time, the data of Mendel and Métais were questioned because of uncertainties in the less precise analytical methods employed. However, the lack of interest in the paper was most likely due to the lack of knowledge and understanding of DNA at this time. Subsequently, the events leading to the current studies of ctDNA required the discovery of the DNA structure and both its relevance to the gene and its appearance in the cytoplasm as both organelle and cytosolic components [3,4].

## 2. Events in the Discovery of DNA Structure and Its Role in the Gene

The concept of Avery et al. [5] demonstrating DNA as genetic material was still being considered. At the same time, Brachet [6], using the methyl green-pyronin reaction and Caspersson [7] employing UV-absorption microscopy and nucleases, as well as the Feulgen reaction were able to demonstrate that both DNA and RNA were present in the nuclei of both plants and animals. Thus, they overturned the idea that DNA was present only in animal nuclei and RNA only present in plant nuclei. This was followed by Alfert and Swift’s [8] demonstration of the fixed amount of DNA per haploid nucleus per species.

An important development concerning the DNA structure was the pronouncement of Chargaff’s rule [9] in which the total number of purines in a DNA molecule equaled the number of pyrimidines. Subsequent X-ray studies enabled the structures of guanine and adenine to be established as well as their mechanism of interaction [10,11]. These results were important for the determination of the structure of DNA [12,13]. The two groups were working on the DNA structure, the former as a theoretical study, the latter by X-ray crystallographic analysis of the purified DNA. An important contributor to the study in Wilkin’s group was Rosalind Franklin who produced a very clear X-ray crystallographic image of DNA. The now famous unpublished photograph found its way to Watson and Crick who then finalized the first DNA model indicating both its structure and replicative ability [12,13]. A game-changing concept was developed by [14,15] with the cell cycle showing DNA to be synthesized at a specific time (S) prior to mitosis (M) with a time gap G1 between M and S, and a second such period G2 between S and M, where G stands for “a gap in our knowledge” (Pelc, personal communication to PBG).

## 3. Cytoplasmic DNA

The identification of cytoplasmic DNA using ultra-violet-light microscopy, biochemistry and autoradiography [15,16,17] was followed by the Chèvremont group [18] showing DNA by the Feulgen reaction to be present in the mitochondria. The concept of cytoplasmic DNA caused great consternation among many biophysicists who insisted that (a) the genes were on chromosomes and the chromosomes were in the nucleus and (b) the genes were comprised of DNA. Therefore, all of the DNA must be in the nucleus [19].

They ignored the fact of cytoplasmic inheritance in plants that was originally thought to involve plastids [20,21].

The idea of DNA mobility was even more abhorrent to many workers who were in the process of establishing the concept of the gene being comprised of DNA.

## 4. DNA Mobility

The concept of the mobility of DNA and also the fact that it could act as a messenger was suggested by Gahan and Chayen [17]. However, at the same time, the experiments of [20,21] also led to the idea of DNA movement. Based on these studies, they were able to further develop the concept of circulating DNA. This began with the repetition of the experiments of the USSR scientist Glouchtchenko [22] who demonstrated the transmission of hereditary characteristics by grafting between two varieties of plants: a mentor plant and a pupil plant.

In 1963, Stroun et al. [20,23] performed similar grafting experiments using egg-plants—*Solanum nigrum* and two varieties of *Solanum melongena*, e.g., *S. melongena* and *S. nigrum*—in which either the stock or the scion was deprived of all growing leaves with a view to subjecting them to the influence of the metabolism of the leaf-bearing section. The products of the pupil plant sometimes showed genetically modified characteristics that were similar to those of the mentor plant. They were strikingly different to those observed upon the sexual crossing of the two varieties. Thus, (a) although some characteristics of the mentor plant were similar to those observed in the pupil plant, others were different to those observed in the mentor plant; (b) the modified pupil plants acquired various mentor-plant characteristics, demonstrating either one or several or all of the characteristics of the mentor; (c) during segregation, which could occur as early as the F1 generation, a few of the recessive parents produced offspring that had dominant features and (d) occasionally, linked characteristics in the mentor plant could be seen individually in the pupil plant and its offspring. Grafting between S. *melongena* and *S. nigrum* yielded similar results. The data were explained as the result of DNA passing from the mentor to the pupil. Hirata [24], also working with *S. melongena,* obtained similar results, concluding that genetic material moved between the stock and the scion.

Similar experiments performed by Yagishita [25,26] employed different species, *Capsicum baccatum* and *Capsicum annuum.* The obtained results also included the non-Mendelian segregation of new features appearing in the graft progeny, as seen in the above studies. Kasahara and co-workers showed that non-Mendelian inheritance also occurred with grafts of *Capsicum annuum* (cited in [27]).

Furthermore, graft-induced genetic variation also occurred upon the transfer of male sterility from male sterile petunia stocks to normal fertile petunia scions [28].

Such preliminary experiments indicated a possible expression of DNA, transferred via the graft, in a subsequent generation (reviewed [29]). These plant experiments were paralleled by a number of similar studies on animals by various researchers [29]. Thus, inspired by the results of Michurin [30,31] and other Soviet researchers on plant graft-hybridization data, Sopikov (1950) determined the changes induced by blood transfusion on hereditary traits. The repeated blood transfusion of 2.5–3 mL blood per kg twice weekly for ten weeks was performed from Black Australorp roosters to White Leghorn hens. The mating of such hens with White Leghorn roosters yielded progeny having a modified inheritance, with some of the progeny having 8–40 black feathers per bird amongst the white plumage. A reciprocal experiment was made with White Leghorn donors and Black Australorp recipients, yielding some progeny with 5–25 white feathers per bird amongst the black plumage. Compared with purebred controls, such progeny also showed an increased body mass and size, as well as longer legs. Furthermore, such studies by Sopikov (1954) injected the blood of Chuvash geese into either White Leghorn or Khaki Campbell ducks, or injected the blood of bronze turkeys into White Leghorn, similarly resulting in abnormal characteristics appearing in the progeny. This approach was used by Sopikov (1966) in order to avoid any adverse effects of inbreeding and to develop new breeding groups, which he did over the subsequent years (Sopikov, 1967, 1980). These results were eventually confirmed by a number of researchers in the USSR as described and summarized in an excellent review by Liu [31]

Western European workers also confirmed these results. For example, Stroun et al. [21] used repeated injections of blood from the grey guinea fowl into White Leghorn variety birds. The progeny showed some grey or black-flecked feathers in the second and later generations. In addition, studies on Rhode Island Red fowls that were repeatedly injected with blood from guinea fowl [32,33,34,35,36] confirmed the initial results of the Soviet workers and Stroun et al. [21].

Such preliminary experiments indicated a possible expression of DNA, transferred via the graft, in a subsequent generation (reviewed by [29]).

These plant experiments were paralleled by a number of similar studies on animals by various researchers [29]. Stroun et al. [21] used repeated injections of blood from the gray guinea fowl into White Leghorn variety birds. The progeny showed some gray or black-flecked feathers in the second and later generations. During this time period and earlier, many such experiments were performed in the USSR [37] in the same time period yielding similar outcomes.

The apparent mobility of DNA was also noted by Gahan and Chayen [17] with a cytoplasmic fraction that appeared to be capable of moving into the nucleus. Overall, the apparent DNA mobility led to experiments being run in order to determine if cells were able to release DNA into their local environment, and if free DNA could be taken up by cells and tissues without degradation. If so, what changes, if any, could such DNA induce in the recipient cells/tissues?

DNA circulation and uptake was demonstratable in both plants and animals. ^3^H-DNA was isolated from thymine-deficient *Escherichia coli* and injected into mice. The radioactive DNA was found in ovarian tissues and especially in the oocyte nuclei. Confirmation was achieved through both CsCl centrifugation and autoradiography [38].

In plants, uptake of DNA into nuclei, mitochondria and plastids of the epidermis, cortex and vascular tissues occurred when cut shoots of *Solanum esculentum* were fed with *E. coli* ^3^H-DNA. Again, the *E. coli* ^3^H-DNA presence was confirmed by both CsCl centrifugation and autoradiography [39,40,41,42]. It was at this point that the study by Koffler et al. [3] indicated increased DNA levels occurring in the blood of lupus erythematosus patients. By 1977, there were many studies indicating the release and uptake of DNA (reviewed by Stroun et al. [4], yet many people were not convinced by the data. Thus, one referee rejected a paper by Gahan, remarking “you should forget the DNA work and concentrate upon the more readily resolvable problems of cell biology”, whilst Stroun and Anker were subject to political attacks and accusations of fraudulent results leading to their failure to obtain research grants [43].

More recently, with the discovery of exosomes, a new DNA carrier exists in blood [44]. Exosomes have been identified as carriers of especially RNAs and proteins [45] between healthy cells, between tumor cells and in both directions between cancer and healthy cells. There is some discussion as to whether or not DNA is also carried by these vesicles [46]. Genomic DNA has been shown to be present [47], and some workers argue that the DNA is attached to the outer surface of the exosomes rather than being inside them. Both whole active mitochondria as well as damaged mitochondria and mitochondrial components—and hence, mitochondrial DNA—have been identified as being present in exosomes [48,49].

## 5. Major Impact Studies

Nevertheless, Leon et al. [50] showed increased DNA levels in the blood of cancer patients. This stimulated a number of studies of which two have been of great importance. The first, in cancer diagnosis, occurred when Stroun et al. [42,51,52] demonstrated the presence of cancer-derived DNA fragments in the blood from patients with a variety of cancer types. This encouraged the search for DNA markers for specific cancer types not only from plasma and serum, but also from other bodily fluids including urine, milk, sputum, saliva, cerebro-spinal fluid, peritoneal fluid and bronchial lavage.

The second important finding was that of Lo et al. [53] in which it was shown that the blood of pregnant women contained fetal DNA fragments. This was followed by the sequencing of this DNA to reveal the genome-wide genetic and mutational profile of the fetus [54]. These results led to the development of a non-invasive, standard technique as opposed to the invasive amniocentesis that can result in fetal death. It has become the method of choice to check human embryos in the first trimester for genetic abnormalities as well as permitting fetal sex and Rhesus status determination [55]. Currently, it is readily available in France and Germany as well as in the UK through the National Health Service [56].

One factor remained in that upon ultra-centrifugation of the cell-culture medium, even at 300k rpm, a newly synthesized DNA remained in the supernatant fraction. This was accompanied by newly synthesized RNA and protein. Efforts to remove the DNA by either centrifuging down a cesium chloride gradient or up a sucrose gradient failed. The outcome was the proposal that the newly synthesized fractions were released from cells as a complex that has been termed a virtosome [57].

Thus, subsequent to the year 2000, the emphasis has very much been on the search for specific markers of various cancer types. Nevertheless, in addition to cancer and exercise studies, the use of cfDNA (cell-free DNA) has been extended to a broad range of clinical studies including multiple sclerosis, cardiovascular disease, stroke, sepsis, hemodialysis, liver and kidney diseases, pancreatitis, tissue transplantation and trauma [58].

## 6. Methodological Development

Such investigations on tumor-derived DNA fragments in blood have led to the need not only to develop analytical methodology to purify and identify the relevant DNA fragments, but also to determine the ideal method of blood drawing, the relevant type of containers to safeguard the drawn blood, blood storage, and the preparation of serum and plasma from drawn blood in order to maintain high-quality cfDNA fragments [59]. Such fragments can be isolated and sequenced.

Some examples of techniques for analyzing cfDNA are shown in Figure 1 and have been detailed and described by Volik et al. [60].

## 7. Evolution of Sequencing

Fifteen years after the discovery of the double helix [12,13] in 1953, the first sequence of a short DNA molecule was published by Wu and Kaiser who analyzed the structure and base sequence at the cohesive ends of bacteriophage lambda DNA [61]. Due to the length of DNA molecules, it was difficult to sequence them. The detection of type II restriction enzymes in 1970 finally prepared the way for successful DNA sequencing [62,63]. This made it possible to cleave large DNA molecules into several smaller fragments in order to subsequently separate them by size by gel electrophoresis. These enzymes recognize specific short nucleotide sequences and cleave them at 4–6 bp in length. The specifically generated ends of these cleaved DNA fragments served as start sequences for DNA sequencing as has occurred in subsequent years.

As reviewed in detail by Hutchison [64], early studies used techniques similar to those used for RNA sequencing. For example, these methods employed either base-specific depurination or *E. coli* (Escherichia coli) nuclease IV, to generate DNA fragments in the range of 10–20 bp at lengths that were separated by either chromatography or electrophoresis. Using these elementary methods resulted in the determination of the operator sequence from the *E. coli* lac operon [65] and a repressor binding site from phage lambda [66].

A further step towards modern DNA sequencing was the plus-and-minus technique developed in 1975 by Sanger and Coulson [67]. In this system, DNA polymerase begins synthesis from a DNA primer by incorporating radio-labeled nucleotides. This labeled product is divided into eight aliquots and used to prime a second round of DNA polymerase reactions. The plus reaction only employs a nucleotide of the four nucleoside triphosphates, thus resulting in all extensions ending with this base. In contrast, the minus reaction uses three of the four nucleotides and produces sequences up to the position before the next missing nucleotide. Following electrophoresis of the eight reactions, a sequence of about 50 bases could be deduced from the developed film.

In 1977, modern DNA sequencing began with the synthesis of the complete DNA sequence of phage ϕX174 by Sanger et al. [68]. These researchers developed further their plus-and-minus method by establishing the dideoxy method [69]. At the same time as the publication of the Sanger dideoxy method appeared, Maxam and Gilbert [70] produced a DNA-sequencing method that was similar to that of Sanger et al. Their method uses a radio-labeled, double-stranded DNA restriction fragment that is cleaved by base-specific chemical reactions. In contrast, Sanger sequencing is based on the random incorporation of chain-terminating dideoxynucleotides by DNA polymerase during DNA replication. For over 40 years, the Sanger technique became the most widely used DNA-sequencing method, and, ten years later, was first commercialized by Applied Biosystems. Eventually, dideoxy sequencers, such as the ABI PRISM, developed by the Leroy Hood group [71] and produced by Applied Biosystems, allowed simultaneous sequencing of hundreds of samples. This approach was employed in the Human Genome Project [72]. The currently used unmodified Sanger method still employs a differently labeled primer in each of the four dideoxy sequencing reactions. After electrophoresis in a single polyacrylamide gel tube, DNA molecules are detected by fluorescent signals as they pass across a detector. The four bases are distinguished by their different dyes that allow deduction of the base sequence. For his contribution to establishing a sequencing method that permits a quick and relatively easy DNA sequencing, Frederick Sanger was awarded a second Nobel Prize in Chemistry in 1980, the first in 1958 for the sequencing of the 51 chain-like amino acids in insulin in 1955 [73]. Such findings advanced sequencing and led to the publication of the three-billion-base sequence of the human genome [72,74]. Sanger DNA sequencing has now been replaced by next-generation sequencing (NGS). However, the Sanger method remains the method of choice for smaller-scale projects.

Different NGS technologies emerged between 1994 and 1998 and have been commercially available since 2005. The common feature of these different procedures is that they are massively parallel, meaning that the number of sequences read in a single experiment is significantly higher than those obtained from capillary-electrophoresis-based Sanger sequencers. They use miniaturized and parallelized platforms for the sequencing of 1 million to 43 billion short reads (50 to 400 bases each) per instrument run [75]. The first of the massively parallel methods to be brought to market was developed by 454 Life Sciences and is based on the pyrosequencing technique [76,77], whilst the Solexa technology differs from the 454 methods by using chain-terminating nucleotides [78]. These and further sequencing methods were described in detail by Hutchison [64] and Heather et al. [79].

## 8. Polymerase Chain Reaction (PCR) Evolution

In the early seventies, the Norwegian postdoc Kjell Kleppe suggested the amplification of DNA using two flanking primers, but the idea was not realized and fell out of use. In 1983, Kary Mullis developed the modern PCR technique. His intention was to develop a novel DNA-synthesis method that artificially duplicates DNA by repeatedly duplicating it in multiple cycles using DNA polymerase. In 1993, Mullis was awarded the Nobel Prize in Chemistry [80] for the method of amplifying DNA to generate several millions of copies of a specific DNA region from a low amount of starting material [81]. In 1985, Saiki et al. [82] enzymatically amplified, for the first time, the sequences of the β-globin, being the first to use PCR [83].

Many different technologies have evolved from original PCR. Among others, digital PCR (dPCR) is capable of determining the absolute quantification of the DNA copy number. This is based on splitting a PCR sample into a thousand subsamples with each having either a single or no copy in each subsample. It is either droplet-based or chip-based. In 1999, dPCR was first mentioned by Vogelstein and Kinzler [84]. Using this method, they quantified *ras* mutations in a reaction by partitioning the sample to perform a series of PCRs. However, the method that they described was not new, having been used over the previous decade. It was termed “single molecule PCR” or “limiting dilution PCR” (reviewed in [85]).

In droplet-based digital PCR (ddPCR), the sample is passed through a microfluidic chip causing the portioning into tens of thousands of droplets separated by mineral oil to form an emulsion. Following PCR, the sample is processed by a flow cytometer to count the number of droplets that include PCR products [86].

In chip-based digital PCR (cdPCR), the sample is loaded into silicon chips. Following the thermal cycling, the chip is imaged by fluorescence microscopy to determine the number of wells containing PCR products [87].

The isothermal nucleic-acid amplification, such as loop-mediated isothermal amplification (LAMP) [88] and recombinase polymerase amplification [89] requires no temperature cycling and has the advantage of having a simpler device design.

However, quantitative real-time PCR using a TaqMan or SYBR Green is the most popular and commonly used technique. The TaqMan real-time PCR [90] uses a TaqMan probe which is a fluorescent DNA probe and is based on the 5′ to 3′ exonuclease activity of *Taq* polymerase. The oligonucleotide probe, with a reporter fluorescent dye labelled at its 5′ end and a quencher dye labelled at its 3′ end, hybridizes to its target gene. During PCR amplification, the quencher dye is cut by the 5′ nuclease activity of *Taq* polymerase, leading to the release of the fluorescent dye. In contrast, the SYBR Green assay [91] uses the SYBR Green I dye, which specifically binds to double-stranded DNA, enabling the detection of products accumulating during the PCR. This assay is simpler and less expensive than the TaqMan but does not use a probe specific to the nucleotide sequence of its DNA target [81].

To date, the development of PCR that delivers high concentrations of pure DNA facilitates DNA sequencing. For example, the Illumina platform uses a modified PCR technique to prepare clusters of single-molecule DNA templates each containing approximately 1000 DNA copies. The Ion Torrent technology discovers protons released as nucleotides and incorporates them during DNA synthesis. For sequencing, the DNA fragments are linked with specific adapters and then amplified by emulsion PCR on the surface of 3-micron-diameter beads [92].

## 9. The Current Status of the Use of cfDNA

### 9.1. In Cancer

The main reasons that the tumor DNA fraction of cfDNA has not been routinely applied in clinics are the low abundance of ctDNA in the pool of cfDNA and the fragmentation of cfDNA in cancer patients [93]. DNA is released into the bloodstream by different sources, including the primary tumor, circulating tumor cells (CTCs), micrometastatic deposits and normal cell types, such as hematopoietic and stromal cells [94]. Thus, both tumor and normal cfDNA circulate in the blood of cancer patients. The amount of ctDNA present in blood may vary due to either the size of the primary tumor or the presence of metastases. Thus, for example, a patient with a tumor weighing 100 g and corresponding to ca. 3 × 1010 tumor cells may release up to 3.3% of tumor DNA into the blood circulation daily [95]. Besides, the DNA fragmentation caused by its primarily apoptotic origin and digestion by DNases impedes its analysis [48,50].

However, over the past decade, advances in the development of ctDNA-detection methods have revolutionized the diagnosis and treatment of cancer. These methods include real-time PCR, dPCR, ddPCR, and NGS-based sequencing [96], which also includes Tagged-Amplicon deep sequencing (TAM-Seq) [97]. They have become very sensitive, being able to assess the low amounts of fragmented ctDNA in the blood of cancer patients with decreased detection errors and background noise. In particular, ctDNA analyses using NGS cancer gene panels have been demonstrated to have the potential to increase the access of ctDNA assays to clinical trials [98].

Below, only a few examples of studies are mentioned reporting recent advances and regarding the multiple clinical applications of ctDNA in the field of precision medicine for cancer patients.

The application of these cfDNA assays in clinical settings, and their ability to identify common mutations and gene fusions may guide personalized targeted therapies [99,100]. In particular, analyses of ctDNA are informative when no tumor tissue is available. Since the extraction of cfDNA occurs in real time, its assessment permits therapy-associated modulations by checking the response to treatment. To date, the relevance of ctDNA testing to guide targeted molecular therapy has also been widely evaluated in non-small-cell lung cancer (NSCLC) patients with epidermal-growth-factor (EGF)-receptor mutations. The ENSURE study showed that plasma EGF-receptor mutations were useful in detecting patients who would benefit from erlotinib treatment [101]. This led to the subsequent US Food and Drug Administration (FDA) approval of the Cobas EGFR Mutation Test v2 (Roche Diagnostics) as the first liquid biopsy test and as a companion diagnostic tool to guide treatment settings.

In addition, the assessment of PIK3CA (phosphatidylinositol 3-kinases) mutations in ctDNA is already an established predictive biomarker. The PIK3CA mutations are attractive therapy targets, so that in May 2019, the FDA approved the drug alpelisib in combination with fulvestrant as a second-line therapy for postmenopausal patients with hormone-receptor-positive, HER2-negative, and PIK3CA-mutated metastatic breast cancer [102]. In patients with metastatic breast cancer, estrogen-receptor-1 (ESR1) mutations in ctDNA are also critical circulating biomarkers. They are frequently subclonal and appear later during metastatic aromatase-inhibitor therapy. In the SoFEA trial, ctDNA analysis showed that ESR1 mutations detected in plasma are useful to direct the choice of further endocrine-based therapy [103].

As reviewed in detail by Ignatiadis et al. [104], the FDA has approved several additional single-gene as well as multigene assays to detect mutations in ctDNA for use as companion diagnostics for specific molecularly targeted therapies for cancer.

For cancer diagnosis, blood tests seem to be useful in discriminating cancer patients from healthy controls and to allow for the detection of early cancer types [105,106]. For example, the detection of circulating, cancer-derived Epstein Barr Virus (EBV) DNA in plasma can be a useful screening technique for nasopharyngeal carcinoma in asymptomatic subjects [107].

In clinical settings, the molecular basis of the acquired resistance to targeted therapies is one of the main challenges. Serial ctDNA analyses have emerged as promising assays to identify acquired resistance and its underlying mechanisms of action. Multiple genomic alterations in genes, such as ESR1, genes of the mitogen-activated protein kinase (MAPK) pathway as well as RB1, KRAS and BRAF that are caused by various therapies have been detected in ctDNA [108].

Increasing evidence suggests that ctDNA may be a predictor of relapse risk. For example, the occurrence of ctDNA in follow-up samples was associated with future recurrence of breast cancer [109]. Moreover, nonmetastatic colorectal cancer patients with a positive ctDNA profile had a recurrence of three months before radiologic or clinical evidence and an incidence of 77%. With a median follow-up time of 49 months, none of these patients without a ctDNA profile had a recurrence [110]. Therefore, ctDNA may be utilized to identify early-stage cancer and predict recurrence.

### 9.2. In General Medicine

The potential clinical utility of cfDNA also plays a role in general medicine [58], ranging from the non-invasive monitoring of infections to the early diagnosis of graft rejection after solid-organ transplantation [111,112], characterization allogenic bone grafting [113], and detection of fetal aneuploidy in pregnant women [114].

In particular, the advancement of PCR and NGS have allowed the detection of low levels of cfDNA from a background signal mixture in physiological conditions and benign diseases [85]. One of the most relevant discoveries for applying cfDNA was identifying fetal cfDNA in maternal blood [53], leading to genetic assays in prenatal diagnostics. The percentage of fetal cfDNA only represents a minor fraction of 3–25% of the total cfDNA level but increases with gestational age and body-mass index, and better detection is possible at about ten weeks of pregnancy [115,116].

As detailed in a review by Polina et al. [117], cfDNA is a diagnostic and prognostic marker for cardiovascular diseases. Thus, uncontrolled hypertension is an independent determinant for elevated cfDNA levels. Furthermore, a multimarker cfDNA test may complement creatine kinase and troponin testing to assess myocardial infarction and ischemic heart failure.

Allograft rejection is one of the major post-transplantation complications affecting graft outcome and survival. The analysis of donor-derived cfDNA in blood serves as a potential tool for early detection of allograft rejection [118].

Neurodegenerative diseases, such as Alzheimer’s, Parkinson’s, Huntington’s diseases, Friedreich’s ataxia, and multiple sclerosis cause the progressive loss of neurons from the nervous system. Nuclear factor-erythroid 2-related factor 2 (Nrf2) is a transcriptional master regulator that supports the redox homeostasis in cells by provoking expression of antioxidant, anti-inflammatory and cytoprotective genes. The analysis of Nrf2 cfDNA can be applied in therapeutic strategies to treat neurodegenerative diseases [119].

Concerning the high prevalence of bacterial and viral infections worldwide, cfDNA is also the most eligible for their diagnosis and prognosis. The identification of such cfDNA and any byproducts can be achieved by PCR and NGS in cfDNA, and is important for a better understanding of their pathogenesis [120].

## 10. Future of cfDNA

The liquid biopsy was originally confined to plasma or serum (Crowley et al., 2013), but has now been expanded to include fluids such as saliva, sputum, urine, breast milk, peritoneal fluid, bronchial lavage, cerebro-spinal fluid and seminal fluid. The additional biopsy systems are of importance since they generally contain larger amounts of ctDNA than are found in plasma/serum. They are becoming increasingly involved in the serial profiling and individualized management of malignant and benign diseases. At present, clinically approved cfDNA assays based on genetic alterations and levels serve as companion diagnostics that facilitate therapy guidance. However, mutation analysis requires a comparatively large amount of cfDNA and therefore high cost for analyses and technical platforms. Test costs vary as a function of the test involved, ranging from £275–£2000 for NIPD tests (plus an additional cost for counselling [121]. Price comparisons of CTC samples depend upon the method employed, with BEAMING costing €486–821 whilst ddPCR costs were €39–298 per sample [122].

The introduction of automation has advanced the possibility of cf/ctDNA-analysis inclusion into routine hospital use with the production of a number of automated systems. Such systems can have minimal human contact periods and turn-around times as low as two hours. However, at present, costs for purchase and maintenance are high and the range of validated tests are low, making the hospital use of such a machine a low priority.

Such automated systems range from those simply determining fragment-size selection of cf/ctDNA to improve the accuracy of next-generation sequencing [123] to those completing the isolation and application of cf/ctDNA in the characterization of relevant disorders linked to specific ctDNA fragments, e.g., AVENIO ctDNA Targeted Kit, the IDYLLA^TM^ platform for mutations in KRAS, BRAF and NRAS and the Agena liquid biopsy mass array system that is currently only for research use. The major application, to date, for such equipment concerns fetal screening with specific systems available to determine trisomy 13, 28 and 21, as well as fetal sex [124,125].

At present, predicting the presence of a primary tumor in advance, as well as knowing when to commence screening, has proved to be extremely difficult, with few methods yielding 100% specificity and sensitivity. This has been exemplified in the mathematical study of cfDNA that can also predict tumor size [126].

Such automated systems will need to be expanded and linked to artificial-intelligence (AI) systems, e.g., machine learning [127]. Its specific ability to identify defined disease signatures will be key to the molecular information achieved from microchip-based diagnostics [128].

However, there are still several further challenges that must be met prior to such machine-learning clinical translation of cfDNA analyses. To date, numerous proof-of-principle studies have shown that there is still a lack of validation studies in larger multicenter clinical studies. In addition, the low level of ctDNA often present in plasma/serum is camouflaged by non-tumor cfDNA that is mainly derived from leukocytes, although this can be improved in some cases by the use of the alternative liquid-biopsy sources. Therefore, techniques such as ddPCR and NGS that target specific circulating molecules may advance the introduction of cfDNA into the clinic. However, extensive analyses and bioinformatic expertise will be required to identify disease-specific markers and to avoid detection errors and background noise.

## 11. Ethical Considerations

The extensive development and use of cfDNA in clinical diagnosis raises a number of ethical questions. Some of these have already been addressed for the application of NIPD [129]. However, this will become even more complex with the advent of next-generation sequencing linked to AI and will result in the ability to identify a series of mutant genes for any individual. This raises a number of questions that are only now beginning to be considered and acted upon.

Some of the mutations will be known to be linked to the development of a clinical disorder. Will the patient eventually suffer from one or more of these disorders?

Should the patient be made aware of such possibilities, e.g., in the case of BBRC1 and BBRC2 and breast cancer? In this case, that patient will have the possibility to either wait and see or accept the traumatic mastectomy. In other situations, the options will not be so clear-cut. The question also arises as to when such analyses should occur given the fact that mutations can be induced throughout the life of an individual.

In a more general situation, a large amount of clinical information will be derived for each individual. How will this information be stored in a safe fashion and who will have access to the data? In addition to the possibility of affecting the patient’s well-being by being clinically screened throughout life, there is the downside that such information could be used to judge if a person is fit for a particular form of work. How will such information be used in a personal life, e.g., for insurance (life, household, travel, etc.)? Will such data lead to problems in buying a car, house or in getting a financial loan for such purposes?

We urgently need answers to such questions.

## Figures and Tables

**Figure 1 diagnostics-12-01192-f001:**
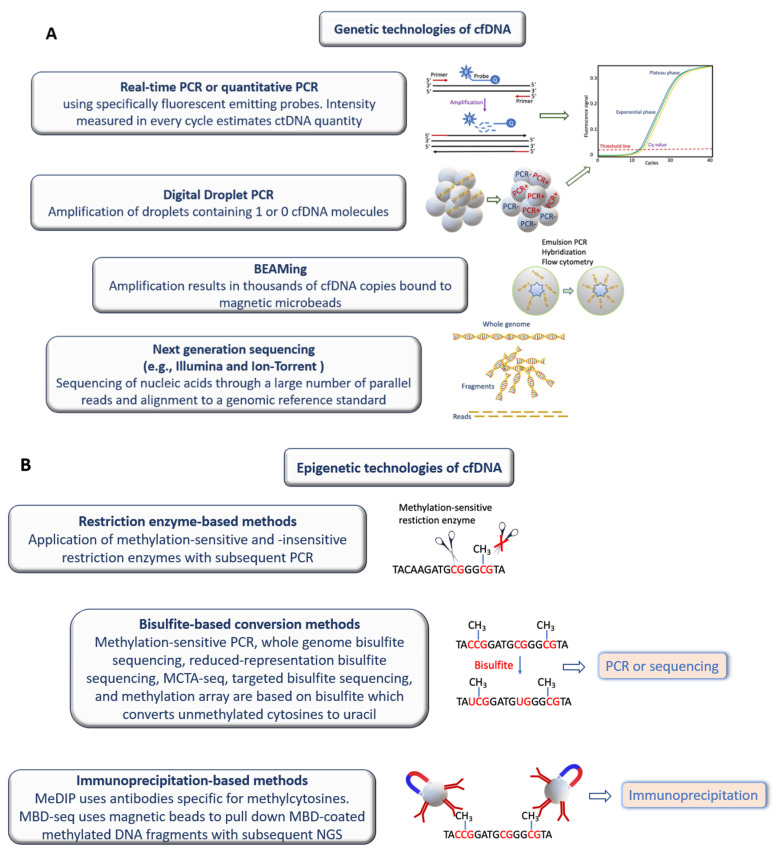
Genetic and epigenetic technologies of DNA. A descriptive summary and pictorial depiction of genetic (**A**) and epigenetic (**B**) technologies of DNA is shown. These diverse techniques were described in detail by Volik et al. [60].

## Data Availability

Not applicable.

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
