# Peer review of "The History and Future of Basic and Translational Cell-Free DNA Research at a Glance"

_diagnostics, 2022, doi:10.3390/diagnostics12051192_

Round 1

Reviewer 1 Report

This manuscript reviews numerous topics relating to circulating DNA. 

The article title is very broad and a bit misleading in terms of the contents of the review article.  There are already numerous reviews of the circulating DNA and applications. It seems this article focuses more on the history of DNA structure and function discovery. A more accurate title should be used.

Section 3 and 4 describe many studies derived from plants. Were any similar studies in human/cell models done? 

It's not clear how section 7 and 8 apply to the history of circulating DNA. Further elaboration is needed to describe how this section relates to circulating DNA. 

Overall, the review needs a better focus on what the contents it intends to review and how it ties into circulating DNA.

Author Response

Comment 1:

This manuscript reviews numerous topics relating to circulating DNA.

The article title is very broad and a bit misleading in terms of the contents of the review article. There are already numerous reviews of the circulating DNA and applications. It seems this article focuses more on the history of DNA structure and function discovery. A more accurate title should be used.

Response 1:

This is not a review, but the first chapter in a book on circulating DNA and the title is that provided by the Editors of the book. We have endeavoured to respond to this title so as to present a backdrop for the remainder of the chapters. However, if the Reviewer feels that the title needs to be revised, perhaps he can offer an alternative that we can discuss with the Editors.

Comment 2:

Section 3 and 4 describe many studies derived from plants. Were any similar studies in human/cell models done?

Response 2:

We thank the referee for raising this point. We have added additional results from the early animal studies. It has to be said that the animal studies were provoked by the results of the plant studies – hence a big mention of the plant experiments.

Comment 3:

It's not clear how section 7 and 8 apply to the history of circulating DNA. Further elaboration is needed to describe how this section relates to circulating DNA.

Response 3:

The historical evolution of the analytical techniques (sections 7, 8) is of importance in the context of this chapter. Without such developments, there would have been no progress in circulating DNA studies, no early analysis of the fetus via amniotic fluid and no use of liquid biopsies now already employed in clinical examinations. Interestingly, Referee 2 specifically cites these sections as of special interest, and has asked us to add a Figure on the techniques discussed that we have prepared as Figure 1.

Comment 4:

Overall, the review needs a better focus on what the contents it intends to review and how it ties into circulating DNA.

Response 4:

As already mention, this is not a Review, but an introductory chapter for a book. As we have indicated in the Abstract, we have followed the indications from the chapter title with the first part of the chapter devoted to the relevant historical aspects of circulating DNA followed by the current use of circulating DNA in clinical situations and ending with possible future developments. 

The English has been corrected as requested.

Reviewer 2 Report

In their mini-review on circulating DNA, the authors have primarily focussed on the methodical analysis of DNA in blood. While this topic has been reviewed prior to this report, the idea of covering these aspects from a historic perspective makes it more interesting and useful for the diagnostics crowd.

The only concern I have is that this report would significantly benefit from the pictorial depiction of some of these technologies for the general readership to be able to appreciate the well-reviewed area. With that covered, I am happy to recommend its publication in Diagnostics.

Author Response

Comment 1:

In their mini-review on circulating DNA, the authors have primarily focussed on the methodical analysis of DNA in blood. While this topic has been reviewed prior to this report, the idea of covering these aspects from a historic perspective makes it more interesting and useful for the diagnostics crowd.

Response 1:

We would point out that this is NOT a mini-review, but an introductory chapter for a book on various aspects of circulating DNA. We are pleased that the referee finds the approach to be of interest.

Comment 2:

The only concern I have is that this report would significantly benefit from the pictorial depiction of some of these technologies for the general readership to be able to appreciate the well-reviewed area. With that covered, I am happy to recommend its publication in Diagnostics.

Response 2:

We thank the referee for the comments and have added Figure 1 covering the methodology, as requested.